# Different individual-level responses of great black-backed gulls (*Larus marinus*) to shifting local prey availability

**Laurie D. Maynard** [ID]*, **Julia Gulka**◉, **Edward Jenkins**◉, **Gail K. Davoren**◉

Department of Biological Sciences, University of Manitoba, Winnipeg, Manitoba, Canada

◉ These authors contributed equally to this work.
* maynardl07@gmail.com

## Abstract

To grow, survive and reproduce under anthropogenic-induced changes, individuals must respond quickly and favourably to the surrounding environment. A species that feeds on a wide variety of prey types (i.e. generalist diet) may be comprised of generalist individuals, specialist individuals that feed on different prey types, or a combination of the two. If individuals within a population respond differently to an environmental change, population-level responses may not be detectable. By tracking foraging movements of great black-backed gulls (*Larus marinus*), a generalist species, we compared group-level and individual-level responses to an increase in prey biomass (capelin; *Mallotus villosus*) during the breeding season in coastal Newfoundland, Canada. As hypothesized, shifts in prey availability resulted in significantly different individual responses in foraging behaviour and space use, which was not detectable when data from individuals were combined. Some individuals maintained similar foraging areas, foraging trip characteristics (e.g., trip length, duration) and habitat use with increased capelin availability, while others shifted foraging areas and habitats resulting in either increased or decreased trip characteristics. We show that individual specialization can be non-contextual in some gulls, whereby these individuals continuously use the same feeding strategy despite significant change in prey availability conditions. Findings also indicate high response diversity among individuals to shifting prey conditions that a population- or group-level study would not have detected, emphasizing the importance of examining individual-level strategies for future diet and foraging studies on generalist species.

## Introduction

With exponential growth of the human population over the last century, wildlife faces reduced availability of pristine habitats and rapidly changing environmental conditions [1]. Individuals must respond quickly and favourably to these anthropogenic-induced changes for populations to persist. Phenotypic plasticity, or the variation in expression of a trait (i.e., physiological, morphological, behavioural) across environmental gradients [2], allows individuals to express traits that maximize fitness under varying environmental conditions [3,4] and in turn can help

**Data Availability Statement:** The data underlying this study are available on Movebank (www.movebank.org) under the study name 'Great Black-

backed Gulls (Larus marinus) Newfoundland' and
the ID number 577076449.

**Funding:** Newfoundland and Labrador Murre
Conservation Fund (19M-3) to GKD https://www.
birdscanada.org/about-us/funding-opportunities/
murre-fund/ National Sciences and Engineering
Research Council of Canada Discovery (2014-
06290) to GKD https://www.nserc-crsng.gc.ca/
Professors-Professeurs/Grants-Subs/index_eng.
asp Ship Time Grants (486208-2016, 501154-
2017, 55517-2018) to GKD https://www.nserc-
crsng.gc.ca/professors-professeurs/grants-subs/
ST_TN_eng.asp#procedures University of Manitoba
Faculty of Science Fieldwork Support program
grants (2016-2018) to GKD. https://umanitoba.ca/
graduate-studies/funding-awards-and-financial-aid
Manitoba Graduate Fellowship to LDM https://
umanitoba.ca/graduate-studies/funding-awards-
and-financial-aid/university-manitoba-graduate-
fellowship-umgf World Wildlife Fund-Canada (G-
0618-583-00-D) to GKD. https://wwf.ca/take-
action/apply-for-funding/ The funders had no role
in study design, data collection and analysis,
decision to publish, or preparation of the
manuscript.

**Competing interests:** The authors have declared
that no competing interests exist.

populations or species buffer against environmental change [5–8]. Alternately, plasticity may be limited, whereby there is a relatively stable expression of a trait by individuals across environmental gradients (i.e. specialization) [9]. Specialization may be favorable in stable environments and help to reduce intra-specific competition, whereby individuals only compete against others with similar trait expressions [6,10], but can also result in lower responsiveness to environmental change at the population level. Therefore, examining plasticity at the population level may mask important individual responses [11]. Studying plasticity at the individual level allows a greater understanding of the response diversity within a population and degree to which it can tolerate varying environmental conditions [12].

A species that feeds on a wide variety of prey types (i.e. generalist diet) may be comprised of generalist individuals, specialist individuals that feed on different prey types, or a combination of the two [13]. Under an increase in prey availability, generalist individuals would modify their foraging behaviour to take advantage of newly abundant resources, while the behaviour of specialist individuals may be unaffected [4,14]. Specialization, however, can be contextual, whereby an individual may be a specialist under stable conditions (e.g., consistent prey availability), allowing resource partitioning and promoting species coexistence, but shifts to a generalist strategy when conditions change (e.g., released from competition, increased prey availability) [5,15]. Alternately, the behaviour of a non-contextual specialist would remain unaffected under shifting environmental conditions [5]. Many large gull species are dietary generalists at the population level [16,17] and prey availability has been shown to affect the diet and foraging behaviour of several gull species. For instance, previous studies have shown that gulls shift their diet to a primarily-fish-based diet, when natural fish prey abundance increases in the surrounding environment [18–20]. Despite known flexibility in large gulls at the population level, individual-level differences in diet and foraging behaviour are also evident in many species [21–24], suggesting that gull populations are comprised of a combination of specialist and generalist individuals that respond to changing prey availability differently [11,25]. Given rapid anthropogenic-induced changes in marine environments, the degree of plasticity in marine predators, including gulls and other seabirds, in response to varying prey resources can have important conservation consequences.

The great black-backed gull (*Larus marinus*), commonly found on the coast of northeastern North America, is considered a dietary generalist species [17]. While great black-backed gulls mainly forage in marine environments by surface-feeding on pelagic fish [26,27], they also kleptoparasitize other seabirds, capture benthic invertebrates in coastal environments, and consume food waste at landfills [17,28,29]. Great black-backed gulls also depredate seabird eggs, chicks and adults, resulting in low breeding performance of other seabird species [30,31]. Most studies of great black-backed gull resource use have focused at the population level [28,32,33], though a few studies have examined inter-individual differences in diet [25] and foraging movements [22,24,34]. The degree to which individual great black-backed gulls alter their foraging behaviour in response to changing resource availability, however, is unknown. In Newfoundland, Canada, a key forage fish, capelin (*Mallotus villosus*), migrates into coastal areas to spawn during the seabird breeding season, causing the inshore prey biomass to double upon arrival at inshore spawning sites [4,35,36]. This dramatic shift in prey availability results in population-level dietary shifts of other seabirds [4,20,21]. For great black-backed gulls, increased capelin availability results in population-level increases in fish prey in chick diet [20] along with reduced predation on other seabirds [30,31,33]. These prey dynamics also provide natural experimental conditions to improve our understanding of individual-level responses of great black-backed gull foraging behaviour and habitat use to changes in resource availability.

The goal of this study is to examine the degree to which studying population-level behavioural responses to changing prey biomass in large gulls can mask important individual responses. To do

this, we compare group-level and individual-specific responses in the foraging movements of great black-backed gulls to a 125% increase in prey biomass (capelin) during July-August, 2018 on the northeast coast of Newfoundland, Canada. We hypothesize that the foraging behavior responses of great black-backed gull to shifting prey (capelin) availability will vary among individuals, owing to inter-individual differences in foraging tactics shown previously for this species [22,25,34], but that a population-level response will be undetectable. Indeed, we predict that changing prey availability will result in divergent habitat use (i.e., marine, terrestrial/coastal, island) and foraging trip characteristics (i.e., trip length, distance from colony, duration) among individuals, whereby these measures may increase, decrease or remain unchanged to varying degrees, while the group-level response will show no statistical differences. As reduced fisheries activities and, thus, reduced supplemental food, have resulted in gull population declines in eastern North America [37,38], understanding individual responses of gulls to changes in natural prey availability will help better understand population-level responses to future changes in fisheries activities [39] as well as shifts in predation pressure on other seabird populations, thereby informing population conservation and management of large gulls and other seabirds in coastal Newfoundland.

## Materials and methods

### Study area and field work

During 2018, we captured adult great black-backed gulls during incubation (n = 7, June 10–11) and chick-rearing (n = 1, July 8, LGM02) from different nests using drop traps on North Cabot Island (49˚10'30.67"N; 53˚21'57.57"W; Fig 1) on the northeast coast of Newfoundland, Canada. This colony is located 1) < 10 km from a cluster of annually-persistent capelin spawning sites [40], 2) < 400 m from a colony of common murres (*Uria aalge*) on South Cabot Island (~10 000 pairs) [40], where gulls have been observed to depredate eggs and chicks, and 3) < 10 km from shore (Fig 1). On the breeding island (North Cabot Island), great black-backed gulls breed among a ~100 pairs of herring gulls (*Larus argentatus*) [41]. A few pairs of black guillemots also breed at the edge of North Cabot Island [41], but no other seabird species were observed breeding on the island. Upon capture, we recorded bill depth at gonys and total head length (mm) to determine sex based on a discriminant function analysis [42], and we deployed solar-paneled GPS loggers (Ecotone® HARRIER-M, ~20g) using a leg-loop harness made of 6.5 mm Teflon tape [43]. Devices weighed between 0.8–1.3% of body mass. GPS loggers recorded latitude and longitude at 15 min intervals and data from loggers were downloaded remotely via UHF to a base station set up near nest sites on the colony. As researchers were not resident on the colony throughout the tracking period, breeding status of nests of tagged individuals were recorded when possible (June 11, July 5, July 21), to account for variation in movement that could be attributed to breeding status and stage.

To determine shifts in prey availability within foraging range of the gull colony (i.e. ~50 km) [22,34,44], we monitored capelin presence at known and persistently used beach (intertidal) and subtidal (15–40 m) capelin spawning sites every 2–5 days throughout July and August 2018 (Fig 1), following [45]. In brief, we monitored spawning sites for evidence of spawning (i.e., presence of capelin eggs, dead capelin, spawning capelin). We considered capelin availability as 'high' when capelin spawning was initiated at one or several spawning sites until capelin spawning had finished and we considered it 'low' outside of this period. Although capelin biomass peaks near the start of spawning [35,36,46], we further corroborated capelin availability periods using published data [36] from a ship-based hydroacoustic survey that quantified capelin biomass ($g/m^2$) over a cluster of subtidal spawning sites (Fig 1). The low capelin availability period included gull movement data both at the start (i.e., incubation) and the end (i.e., chick-rearing) of the study period (July-August).

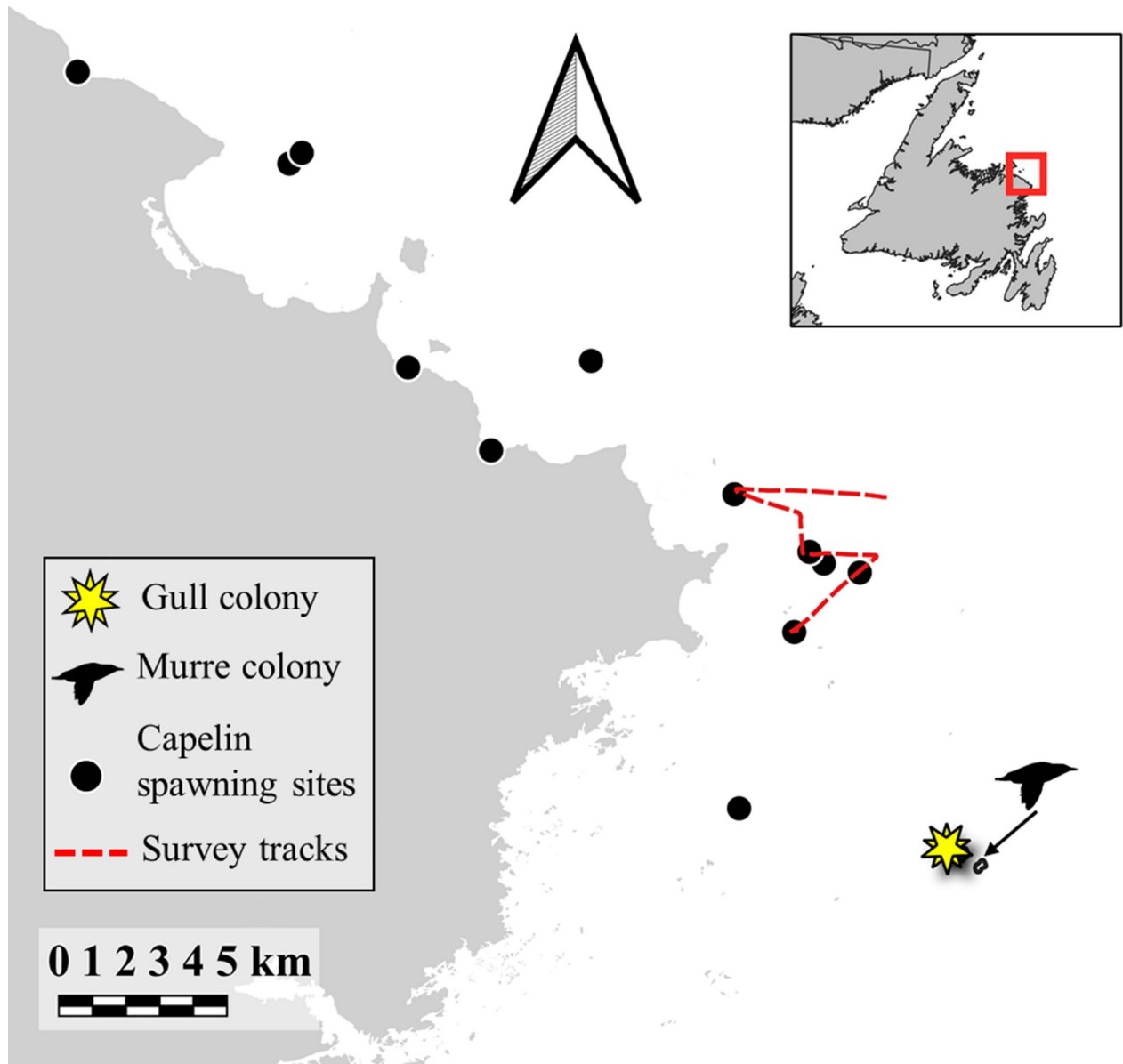

**Fig 1. Location of the breeding colony of GPS-tagged great black-backed gulls (*Larus marinus*; yellow) as well as a nearby common murre (*Uria aalge*) colony, known subtidal and intertidal capelin spawning sites, and the hydroacoustic survey track used to quantify capelin biomass during July-August, 2018 in coastal Newfoundland, Canada.**

## Data analysis

We first quantified GPS locations as either on or off the breeding island using the intersect function in QGIS [47]. To avoid comparing distinct foraging trips with short non-foraging trips, we conservatively defined a foraging trip as trips off the breeding island > 2 hours, as used in other studies [34,48]. Within each trip, we considered point locations under 4 km/h as foraging/roosting locations, following [48] and [34]. Locations are considered foraging/roosting locations because we cannot discriminate between these two behaviours at the temporal resolution locations were recorded (i.e., 15 minute). For each foraging trip, we quantified a

number of foraging trip parameters, including mean distance from shore over the complete trip (km), mean distance from the colony over the complete trip (km), total trip distance (km) and total number of foraging/roosting locations per habitat. During data exploration, we compared models with mean and maximum distance parameters and found that they had similar results, but chose the model with the mean distance as it had the lowest AIC value. We also assigned habitat type to each foraging/roosting location using open-source polygons (open. canada.ca). Habitats included marine (indicating pelagic habitat use), coastal/terrestrial (indicating use of intertidal and anthropogenic habitats), and island (indicating presence at seabird breeding colony, i.e. likely predation).

To compare individual- and group-level foraging responses to prey availability, we examine space use at both the group and individual level. For all statistical tests, we used an alpha value of 0.05. For space use, we calculated the normalized 50% utilization distributions (UD) of GPS locations using the *adehabitatHR* package [49] in R version 4.0.1 [50] at two temporal scales to represent core foraging areas. First, we calculated the 50% UDs at the scale of the individual (all foraging trips combined) and group (all trips from all individuals combined) by capelin availability period (low, high). At the individual scale, we normalized UDs by randomly sub-sampling the GPS locations for each individual to ensure that the number of locations were the same in each capelin availability period. We compared the size of core foraging areas for each individual between capelin availability periods using a paired t-test. Second, we calculated 50% UDs at the scale of each foraging trip by individual and we quantified the spatial overlap of foraging trips using Bhattacharyya's affinity index (BA) [51], where a value of 0 indicates no spatial similarity and a value of 1 indicates complete spatial similarity. To investigate the spatial consistency of foraging space of each individual within a capelin availability period, we calculated the BA between all possible pairs of foraging trips made by an individual during low and high capelin availability periods separately. We then calculated the mean spatial overlap within a capelin availability period for all individuals combined, and then for each individual. These mean BA values represent consistency in foraging space use by each individual or the group within a capelin availability period. We then conducted two analyses of variance on the BA values: (1) individual-level model, with individual and capelin availability as fixed predictor variables, including their interaction; (2) group-level model, with capelin availability only. These ANOVAs allowed us to investigate whether some individuals were more spatially consistent (i.e. high mean BA) within a capelin availability period than other individuals (individual-model), and whether the group (all individuals combined) were more spatially consistent within a capelin availability period. If the predictor was significant, post-hoc Tukey tests were performed.

To investigate the spatial consistency of foraging trips across capelin availability periods, we calculated the pairwise spatial overlap (BA) of foraging trips made by an individual during the high capelin availability period with foraging trips during the low capelin availability period. We then calculated the mean spatial overlap across capelin availability periods for each individual and then all individuals combined. These mean BA values represent consistency in foraging space use across capelin availability periods, whereby a high (scale 0–1) BA would indicate that an individual or the group (all individuals combined) used the same foraging areas in both capelin availability periods. An analysis of variance was then performed on the BA values of each individual, with individual as the fixed predictor variable, allowing us to investigate whether some individuals were more consistent in space use across capelin availability periods (i.e. high mean BA) than other individuals. These mean BA values of each individual where then qualitatively compared to the mean BA of all individuals combined.

We also compared foraging responses to shifting prey availability by examining changes in foraging trip characteristics (i.e. distance and habitat use) between capelin availability periods. We used two general linear mixed models: (1) individual-level model, with capelin (prey)

availability period (i.e. high, low), individual, and capelin availability period x individual as fixed factors and sex as a random effect; and (2) group-level model, with only capelin availability period as a fixed factor and sex as a random effect. In our first model (individual-level model), we included individuals as a fixed effect rather than a random effect, which is often used with repeated measures [52], because our primary goal was to compare responses to shifting capelin availability among individuals. We conducted both models for each foraging trip characteristic (mean distance from shore, mean distance from colony, total trip distance; six models total). For habitat use, we used a similar method where fixed predictor variables were the same as stated above, except that the response variables were square-root transformed proportions of foraging/ roosting points within marine, coastal/terrestrial and island habitat per foraging trip (six additional models, total of 12). For all individual-level models, when the interaction between capelin availability and individual was significant, we used *a priori* contrasts to compare means between individuals to evaluate individual differences between low and high capelin periods.

Research protocols were reviewed and approved by University of Manitoba Animal Care Committee in adherence with the Canadian Council of Animal Care (F16-017/1). All applicable international, national, and/or institutional guidelines for the care and use of animals in research were followed. All procedures performed in studies involving animals were in accordance with the ethical standards of the institution or practice at which studies were conducted. Research was conducted under a Canadian Scientific Permit to Capture and Band Migratory Birds (10873). Anesthesia or euthanasia was not necessary for this study and birds were safely released after handling.

## Results

GPS loggers were deployed on six male and two female great black-backed gulls, which recorded data for 5–65 days (285 foraging trips total) from June 10—August 14 (Table 1). Capelin began spawning on July 10 and remained present at spawning sites until August 10, and, thus, capelin availability was classified as high from July 11—August 9 (89 foraging trips; 8 individuals tracked) and classified as low from June 10 –July 10 and August 10–14 (196 foraging trips; six individuals tracked). Five hydroacoustic surveys (July 9, 18, 24, 28, August 7) corroborated these prey availability periods, with capelin biomass peaking on July 28 (0.239 g/m$^2$; see Berard & Davoren 2020 for details) with a mean density of 0.0138 g/m$^2$ during low capelin availability. Nest failure occurred in six of the initial seven tracked individuals between June 8

**Table 1. Deployment information and foraging characteristics of eight GPS-tracked great black-backed gulls (*Larus marinus*) during June-August, 2018 in coastal Newfoundland.** Deployment information includes deployment date, sex, breeding status, and number of days tracked. Foraging characteristics include number of foraging trips and sizes of core foraging areas (50% utilization distributions) in both low (June 10-July 10, August 10–14) and high (July 11-August 9) capelin availability periods. Note that LMG03 and LMG07 were not tracked during high capelin availability.

| Deployment | | | | | No. Trips | | Core Foraging Areas (km$^2$) | |
| --- | --- | --- | --- | --- | --- | --- | --- | --- |
| Individual | Date | Sex | Status | Days | Low | High | Low | High |
| LMG01 | June 10 | M | Failed | 54 | 28 | 19 | 699 | 593 |
| LMG02 | July 8 | M | Breeding | 5 | 3 | 4 | 279 | 186 |
| LMG03 | June 10 | M | Failed | 11 | 10 | - | 125 | - |
| LMG04 | June 10 | M | Failed | 60 | 44 | 19 | 178 | 539 |
| LMG05 | June 10 | M | Failed | 41 | 13 | 11 | 174 | 621 |
| LMG06 | June 10 | M | Failed | 65 | 58 | 30 | 5.83 | 2.86 |
| LMG07 | June 10 | F | Breeding | 10 | 9 | - | 0.83 | - |
| LMG08 | June 11 | F | Failed | 39 | 31 | 6 | 67 | 43 |
| *All individuals* | | | | | *196* | *89* | *121* | *216* |

—July 7. The tracked individual with a successful nest had two chicks on July 5, but had lost its GPS logger after 10 days (LMG07; Table 1). The GPS logger deployed on July 8 (LMG02) was lost after 5 days during which the nest site had two chicks. Therefore, GPS loggers from breeding individuals (i.e., LMG02, LMG07) recorded for 5–11 consecutive days (7–9 foraging trips per individual) and failed breeders for 12–65 days (10–88 foraging trips per individual, Table 1), though the exact date of nest failure is unknown. Owing to variation in the number of tracking days per individual, we explored the impact of tracking duration by conducting all analyses and calculating utilization distributions on the first 10 days of tracking as well as tracking days 10–20 and all tracking days. As the results did not differ based on the study period used, we present results using all tracking days.

The size of core foraging areas of individuals increased between low and high capelin availability periods for some individuals but decreased for others (Table 1 and Fig 2). Although the group-level core foraging area was 95 km$^2$ smaller during low compared to high capelin availability (Table 1), this difference was not statistically significant (Table 1; $t_{12}$ = 1.03; p = 0.33). When examining the spatial overlap of foraging trips made by individuals within each prey availability period, the individual-level model showed a significant interaction between individual and capelin availability ($F_5$ = 42.87; p < 0.001), and contrasts showed that two individuals exhibited higher spatial overlap of foraging trips during high capelin availability compared to low capelin availability, while three individuals exhibited the opposite pattern (Fig 3). The group-level model, however, showed no significant differences in spatial overlap of foraging trips between capelin availability periods ($F_1$ = 1.67; p = 0.196; Fig 3). When examining spatial overlap of foraging trips made by individuals across capelin availability periods, the amount of overlap differed significantly among individuals ($F_5$ = 53.38; p < 0.001; Fig 4), while group-level overlap was higher (0.27 ± 0.28) than all individuals except one (Fig 4). Although spatial overlap of foraging trips made by individuals across capelin availability periods was generally low, one individual (LMG06) had a significantly higher overlap (mean BA = 0.34) than the other individuals (mean BA = 0.14–0.24; Fig 4).

Foraging trip characteristics and habitat use differed among individuals and between capelin availability periods in the individual-level model, but did not differ between capelin availability periods in the group-level model (Table 2 and Figs 5 and 6). At the individual level, the interaction was significant for all the models, indicating that responses to shifting capelin availability differed among individuals. Three gulls (LMG06, LMG01, LMG08) exhibited different foraging behaviour (Fig 5) and habitat use (Fig 6) under varying capelin availability, while two other gulls (LMG04, LMG05) did not exhibit any significant changes (p-values = 0.07–0.93). Of those exhibiting differences in behaviour, with a shift from low to high capelin availability, one individual (LMG06) increased trip length, utilized areas closer to shore, and decreased use of island habitat relative to coastal/terrestrial and marine habitats. Another individual (LMG01) also foraged closer to shore, and decreased marine habitat use during high capelin availability, but did not significantly increase use of coastal/terrestrial or island habitats. Lastly, a third individual (LMG08) showed an opposite response compared with the other two individuals by foraging farther from shore and increasing island habitat use during high relative to low capelin availability. Responses of two gulls (i.e., LMG03, LMG07) could not be evaluated as they were not tracked across both capelin availability periods.

## Discussion

As predicted, shifts in capelin availability resulted in different individual responses by great black-backed gulls in foraging behaviour and space use, whereas group-level differences were undetectable. Indeed, some individuals maintained similar foraging areas, movement patterns (i.e. foraging trip characteristics) and proportional habitat use under varying capelin

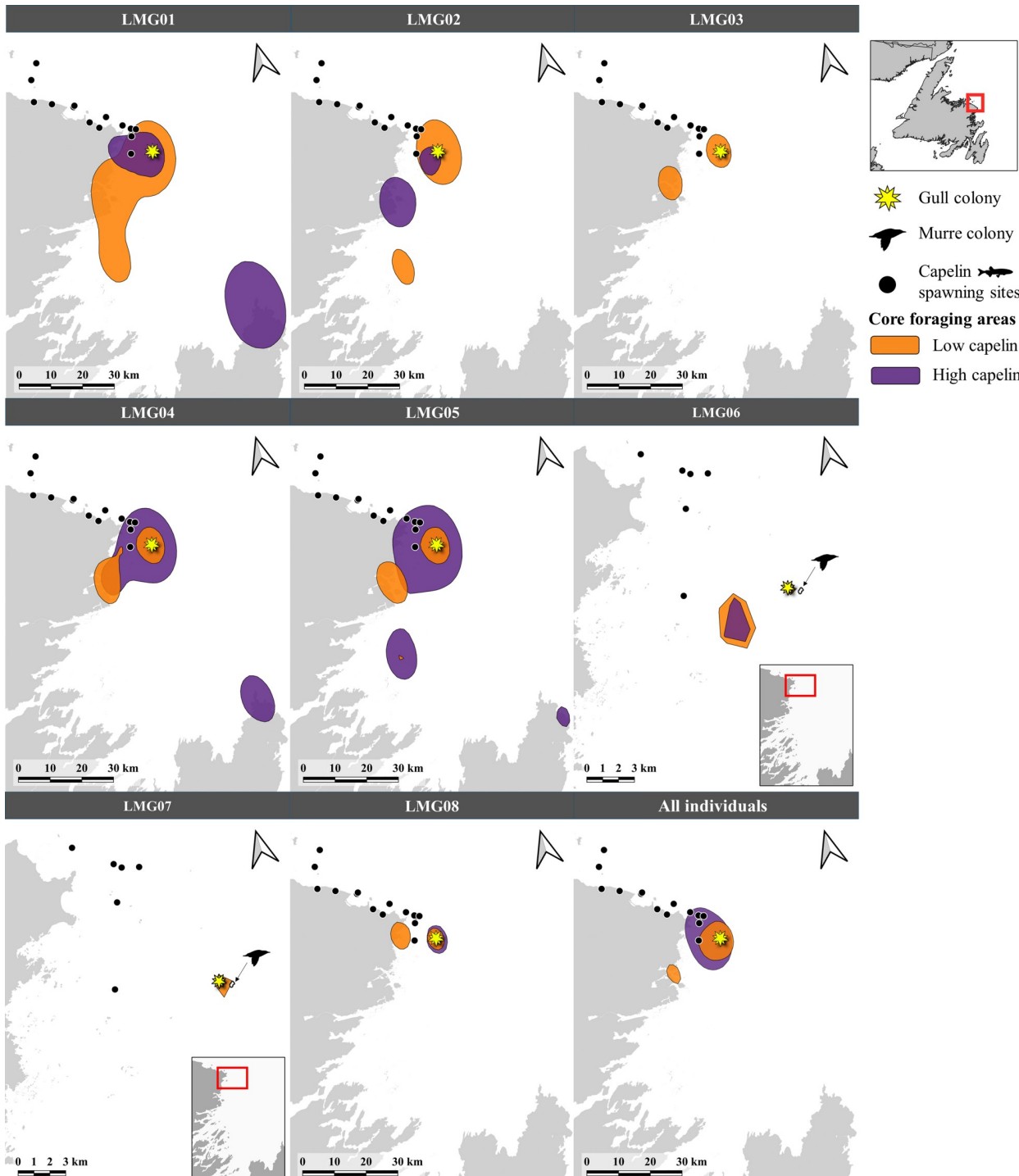

**Fig 2. Core foraging areas (i.e. 50% utilization distributions) for each great black-backed gull (n = 8) as well as all individuals combined, during low and high capelin availability periods in coastal Newfoundland during June-August 2018.** Note that LMG03 and LMG07 were not tracked during high capelin availability, and spatial scale of LMG06 and 07 are different than the other six individuals.

availability, while others shifted foraging areas and either increased or decreased movement and proportional use of certain habitat types. These results suggest the presence of individual

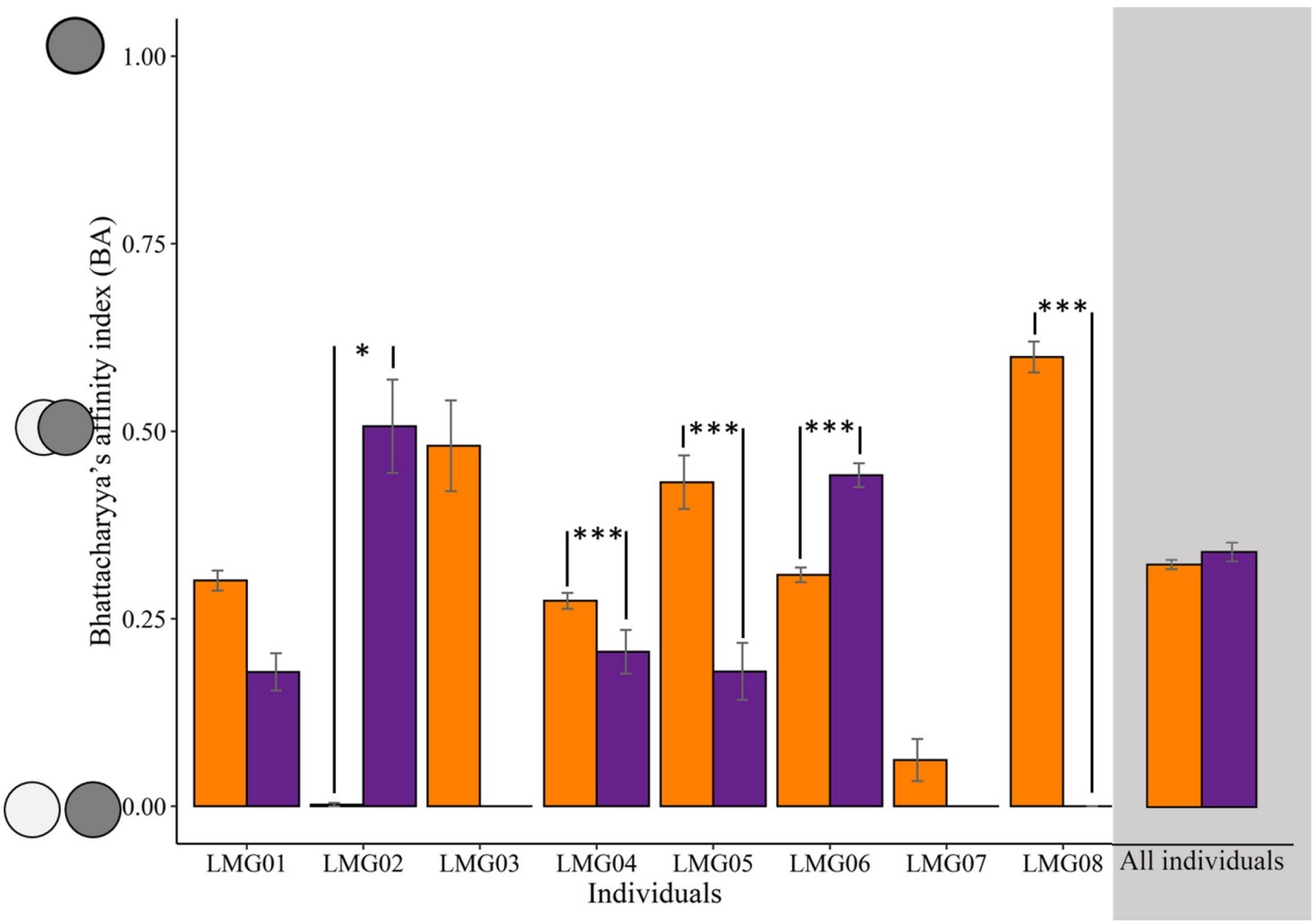

**Fig 3. Mean (± SE) spatial overlap (Bhattacharyya's index) of foraging trips of each individual great black-backed gull and all individuals combined within capelin availability periods (low = orange, high = purple) in coastal Newfoundland during June-August 2018.** Note that LMG03 and LMG07 were not tracked during the high capelin availability period. Asterisks indicates significance level: * p < 0.05; ** p < 0.01; *** p < 0.001.

specialization, both contextual and non-contextual, and that these responses to varying resource availability would have gone undetected if focused at the population level.

As the great black-backed gull is a generalist species in both diet and foraging behaviour [17,26], we expected that some individuals within the population might modify their foraging behaviour under shifting prey regimes [9]. Indeed, individuals appeared to use different foraging areas in each capelin availability period, as evidenced by the low spatial overlap (BA < 0.25) of foraging areas for most tracked gulls across capelin availability periods. Specifically, four of the six individuals (LMG01, LMG02, LMG06, LMG08) altered movement patterns and/or habitat use with the increase in capelin availability, suggesting a switch to a fish-based diet and corroborating previous diet studies [20]. These differences in foraging behaviour were not evident at the group-level, whereby the size of core foraging areas and mean spatial overlap of foraging trips were similar across periods. This lack of group-level response was expected but still surprising as the arrival of spawning capelin in coastal Newfoundland is known to affect population-level seabird diet [20,35,53] including the great black-backed gull [21,30,31]. For generalist and opportunist species, changes in prey availability may have

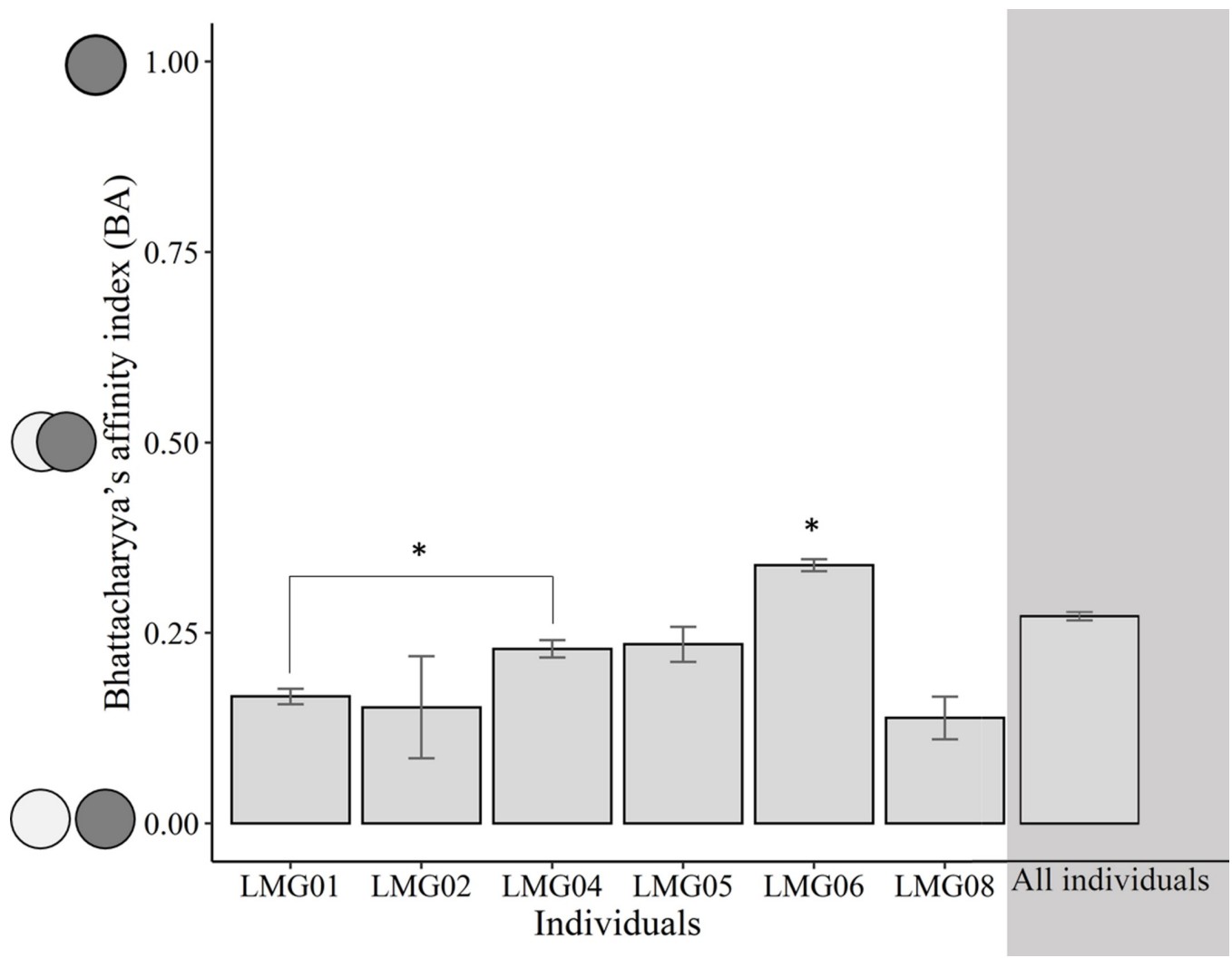

**Fig 4. The mean (± SE) spatial overlap (Bhattacharyya's index) of foraging trips made by individual great black-backed gull across low and high capelin availability periods in coastal Newfoundland during June-August 2018.** Asterisks indicates significance level: * p < 0.05; ** p < 0.01; *** p < 0.001.

**Table 2. Summary results of analysis of variances for foraging trip characteristics, including total trip distance (km), mean distance from shore (km), mean distance from the colony (km), along with the proportion of foraging/roosting locations per habitat type of GPS-tracked great black-backed gulls (*Larus marinus*) during June-August, 2018 in coastal Newfoundland.** Habitat types included marine (pelagic foraging), coastal/terrestrial (intertidal foraging and foraging at anthropogenic sources), or island (seabird colony).

| | Foraging trip characteristics | | | | | | Habitat use | | | | | |
|---|---|---|---|---|---|---|---|---|---|---|---|---|
| | Trip Distance | | Dist. from Shore | | Dist. from Colony | | Marine | | Island | | Coastal/Terrestrial | |
| | *F* | *p* | *F* | *p* | *F* | *p* | *F* | *p* | *F* | *p* | *F* | *P* |
| *Individual-level model* | | | | | | | | | | | | |
| Individual | 6.59 | < **0.001** | 4.2 | < **0.001** | 10.98 | < **0.001** | 13.41 | 0.02 | 22.89 | < **0.001** | 6.02 | < **0.001** |
| Capelin availability | 0.94 | 0.33 | 2.87 | 0.1 | 0.84 | 0.36 | 3.56 | < 0.001 | 0.33 | 0.54 | 2.73 | 0.1 |
| Individual:Capelin | 5.17 | < **0.001** | 4.58 | < **0.001** | 3.39 | **0.005** | 4.34 | < **0.001** | 4.18 | < **0.001** | 2.79 | **0.02** |
| *Group-level model* | | | | | | | | | | | | |
| Capelin availability | 1.78 | 0.18 | 1.93 | 0.2 | 0.1 | 0.75 | 2.55 | 0.11 | 0.2 | 0.65 | 2.52 | 0.11 |

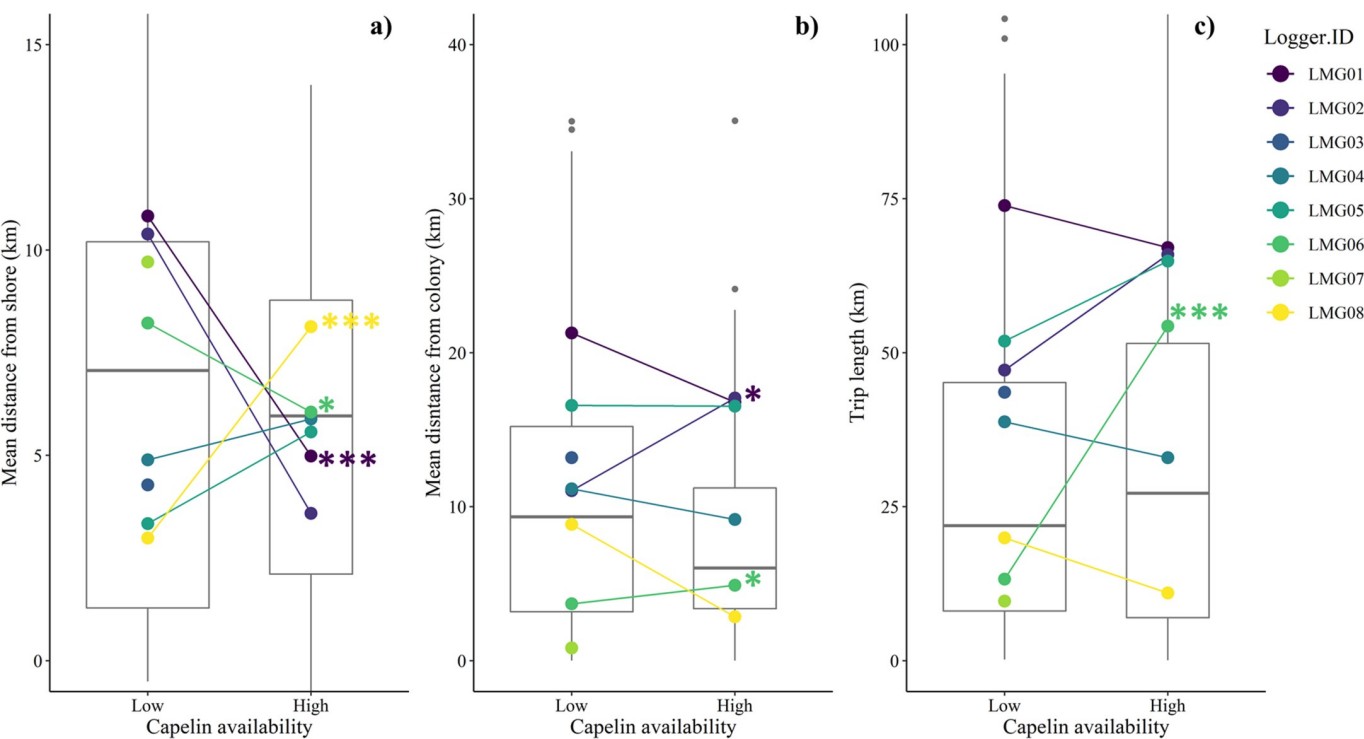

**Fig 5. Comparing foraging trip characteristics between capelin availability periods (low, high) for all great black-backed gull during June-August 2018 in coastal Newfoundland for all individuals combined (boxplot with median indicated by the horizontal line), as well as for each individual separately (mean of each individual indicated by color).** Trip characteristics included a) mean distance from shore (km); b) mean distance from colony (km); c) total trip length (km). Asterisks indicate significance level (* p < 0.05; ** p < 0.01; *** p < 0.001). Note that LMG03 and LMG07 were not tracked during the high capelin availability period and that y-axes differ by panel.

different effects on individuals [11,54]. Indeed, during the high capelin availability period, capelin was readily available at wharfs (from fisheries activities) as well as in coastal and marine habitats where capelin spawn [4,55], and at seabird colonies, where provisioning seabird parents, such as common murres (*Uria aalge*), are targets for kleptoparasitism [21,56]. Although gulls likely all switch to a fish-based diet with increased capelin availability to some degree, fish was accessible from different sources and habitats. Therefore, individuals likely responded differently according to their respective temperaments [14] and responses were disguised when examining at the group-level. While small sample sizes (i.e. number of individuals tracked) may play a role in the lack of group-level patterns observed in this study, we highlight the importance of individual-level differences in behaviour and space use of six individuals that were tracked across capelin availability periods. Although higher sample sizes would likely improve understanding of the amount of individual variation present in the population, we predict that combining data across a larger sample would likely still result in undetectable changes at the group level. Additionally, targeting these individual-level responses highlights the diversity of responses among individuals, which will play an important role in understanding the population dynamics and persistence of this and other generalist populations.

In our study, three of the six individuals tracked during both capelin availability periods showed minimal behavioural and/or spatial response to varying capelin availability, indicative of individual specialization. These individuals had slightly higher spatial overlap across capelin availability periods, as well as similar foraging trip characteristics and habitat use between capelin availability periods. For instance, one individual (LMG06) foraged in the same area during

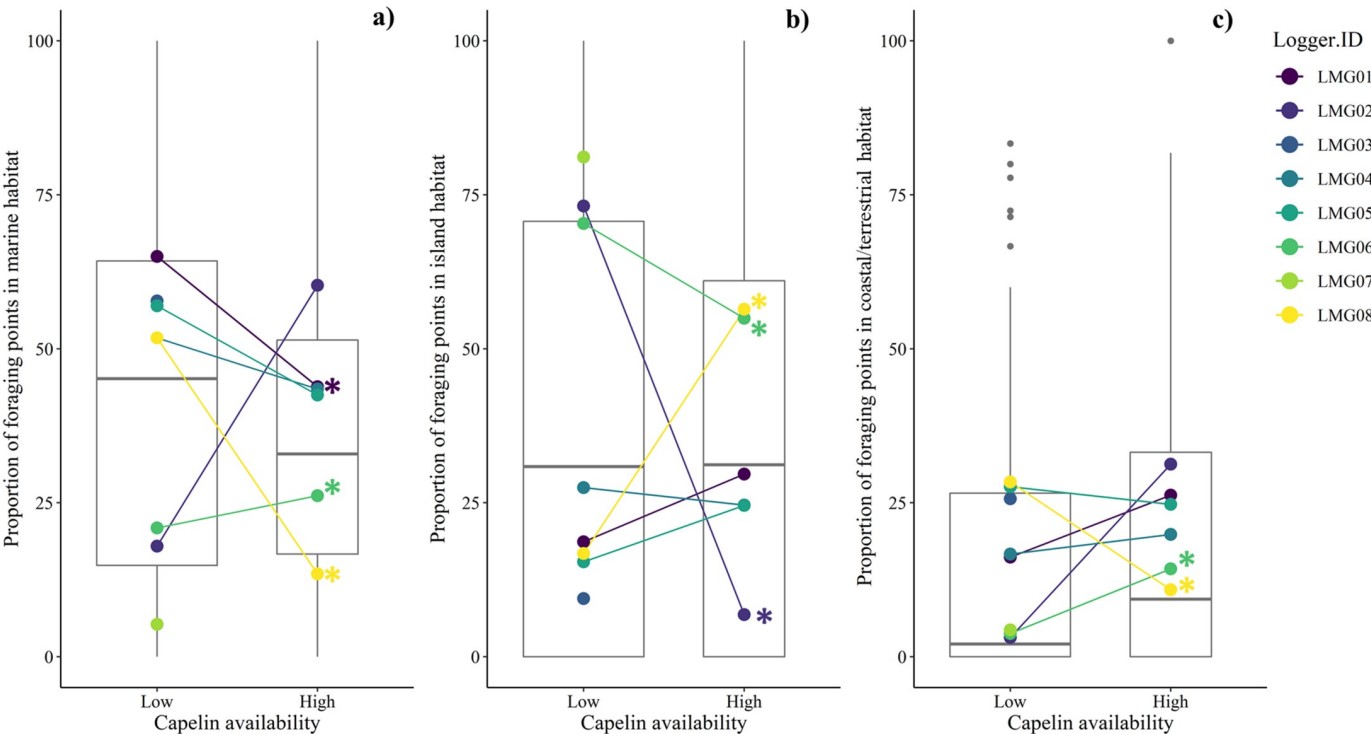

**Fig 6. Proportions of foraging/roosting locations per habitat type by capelin availability period for all great black-backed gull individuals (mean indicated by the horizontal line in the box plot), as well as each individual separately (mean of each individual indicated by color) during June-August, 2018 in coastal Newfoundland.** Habitat types included a) marine b) island and c) coastal and terrestrial habitats. Asterisks indicate significance level (* p < 0.05; ** p < 0.01; *** p < 0.001). Note that LMG03 and LMG07 were not tracked during the high capelin availability period.

high and low capelin availability, illustrated by the smallest core foraging area and the highest spatial overlap across periods. This individual could be considered a non-contextual specialist, i.e., an individual that remains specialized despite changes in prey availability [5]. These findings generally suggest that great black-backed gull populations may be comprised of both non-contextual and contextual specialists. Non-contextual specialists have been observed in other gull species (e.g., dolphin gulls *Leucophaeus scoresbii* [54]; yellow-legged gulls *Larus michahellis* [11]) and are likely competitively dominant individuals in the population, that feed on high-quality prey and protect access to this food resource [57]. When prey availability increases, these specialists may further benefit, such as through reduced resource guarding of high-quality prey [5]. In contrast, contextual specialists may be subordinate individuals that are restricted to a portion of their niche due to competitive interactions [58], but may show plasticity when released from competition [5,15,57]. Similarly, individual generalists may be subordinate individuals that are restricted to a generalist diet, until released from competition, allowing them to specialize on a high-quality prey and become contextual specialists [57,59]. While individual specialization has been reported multiple times in great black-backed gulls [22,24,25], these studies did not quantify gull behaviour under varying resource conditions. Therefore, it is unknown if individual specialization in these previous studies was contextual [5,9]. We show that individual specialization in great black-backed gulls can be non-contextual in some individuals, and suggest future studies should explore the prominence of non-contextual specialization as well as the link between behavioural and dietary specialization.

As the majority of our individuals were male failed breeders, we are confident that most of the behavioural variation is due to an increase in prey availability. We unfortunately lacked the

sample size to infer additional potential sources of behavioural variation, such as sex and breeding status [60,61]. For instance, male lesser black-backed gulls (*Larus fuscus*) travel further from the colony and spend more time offshore than females [62]. Additionally, female black-browed albatross (*Thalassarche melanophrys*) had more variable behaviour and wider foraging ranges than males, which stayed closer to the colony [14]. Breeding status can also affect behavioural consistency [63]. For instance, northern gannets (*Morus bassanus*) with unsuccessful nests had an intermediate fidelity to foraging sites relative to immatures and active breeders [63]. This avenue of research should be further studied in gulls.

## Conclusions

This study provides novel information on foraging behaviour plasticity of individual great black-backed gulls, while exemplifying the potential to mask important individual-level responses when focusing foraging ecology studies solely at the population level. While a small sample size limits our ability to make population-level conclusions on the response of a generalist species to changes in prey availability, we show a diversity of individual responses and therefore highlight the importance of individual-level studies in generalist species. High response diversity allows populations to buffer against environmental change [8], which may explain why gull species persist and often thrive in rapidly changing environments, such as urban habitats [38]. Indeed, different responses to environmental change among individuals may reduce intraspecific competition and, thus, individuals may only compete with others that behave similarly [10,64]. Such a strategy to minimize intraspecific competition may be particularly important in highly competitive species, such as large gulls, which are known to defend food resources and act aggressively, depriving other conspecifics or heterospecifics access to resources [21,28]. Finally, the results of this study exemplify the importance of measuring individual responses to changing resources and/or environmental conditions, to examine the degree to which specialization is contextual. This distinction informs the ecological mechanism of specialization and a population's capacity to adjust to environmental change.

## Acknowledgments

Thanks to all Davoren Lab students and the crew members of the FV *Lady Easton* for support during field work. Thanks to all reviewers for their helpful comments.

## Author Contributions

**Conceptualization:** Laurie D. Maynard, Julia Gulka, Edward Jenkins.

**Data curation:** Laurie D. Maynard.

**Formal analysis:** Laurie D. Maynard.

**Funding acquisition:** Laurie D. Maynard, Julia Gulka, Edward Jenkins, Gail K. Davoren.

**Investigation:** Laurie D. Maynard, Julia Gulka, Edward Jenkins.

**Methodology:** Laurie D. Maynard, Julia Gulka, Edward Jenkins, Gail K. Davoren.

**Project administration:** Laurie D. Maynard, Julia Gulka, Edward Jenkins, Gail K. Davoren.

**Resources:** Gail K. Davoren.

**Supervision:** Gail K. Davoren.

**Validation:** Laurie D. Maynard, Julia Gulka, Edward Jenkins, Gail K. Davoren.

**Visualization:** Laurie D. Maynard.

**Writing – original draft:** Laurie D. Maynard.

**Writing – review & editing:** Laurie D. Maynard, Julia Gulka, Edward Jenkins, Gail K. Davoren.

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
