## [Decision Letter · Decision Letter 0]

14 Jun 2021

PONE-D-21-15967

Different individual-level responses of great black-backed gulls (Larus marinus) to shifting local prey availability

PLOS ONE

Dear Dr. Maynard,

Thank you for submitting your manuscript to PLOS ONE. After careful consideration, we feel that it has merit but does not fully meet PLOS ONE’s publication criteria as it currently stands. Therefore, we invite you to submit a revised version of the manuscript that addresses the points raised during the review process.

We look forward to receiving your revised manuscript.

Kind regards,

Vitor Hugo Rodrigues Paiva, Ph.D.

Academic Editor

PLOS ONE

Journal Requirements:

5. We note that Figures 1 and 2 in your submission contain map images which may be copyrighted. All PLOS content is published under the Creative Commons Attribution License (CC BY 4.0), which means that the manuscript, images, and Supporting Information files will be freely available online, and any third party is permitted to access, download, copy, distribute, and use these materials in any way, even commercially, with proper attribution. For these reasons, we cannot publish previously copyrighted maps or satellite images created using proprietary data, such as Google software (Google Maps, Street View, and Earth). For more information, see our copyright guidelines: http://journals.plos.org/plosone/s/licenses-and-copyright.

You may seek permission from the original copyright holder of Figures 1 and 2 to publish the content specifically under the CC BY 4.0 license. 

If you are unable to obtain permission from the original copyright holder to publish these figures under the CC BY 4.0 license or if the copyright holder’s requirements are incompatible with the CC BY 4.0 license, please either i) remove the figure or ii) supply a replacement figure that complies with the CC BY 4.0 license. Please check copyright information on all replacement figures and update the figure caption with source information. If applicable, please specify in the figure caption text when a figure is similar but not identical to the original image and is therefore for illustrative purposes only.

Reviewers' comments:

Reviewer's Responses to Questions

**Comments to the Author**

1. Is the manuscript technically sound, and do the data support the conclusions?

Reviewer #1: Partly

Reviewer #2: Yes

2. Has the statistical analysis been performed appropriately and rigorously? 

Reviewer #1: Yes

Reviewer #2: Yes

3. Have the authors made all data underlying the findings in their manuscript fully available?

Reviewer #1: Yes

Reviewer #2: Yes

4. Is the manuscript presented in an intelligible fashion and written in standard English?

Reviewer #1: Yes

Reviewer #2: Yes

5. Review Comments to the Author

Reviewer #1: This paper presents a study investigating a unique idea in theoretical ecology and is worthy of publication upon addressing suggested edits. The largest overall comment to this paper is there is no clear link made between capelin biomass and great black-backed gull foraging/diet. Are the authors suggesting that black-backs are eating the capelin? Are they suggesting that increase capelin abundance also increases the abundance of larger fish that the black-backs eat? Creating a clearer picture here would be extremely beneficial to the paper.

Line by line comments:

Line 66: change “should” to “may”

Line 67: it would be beneficial to expand here as to what you mean by “stable” conditions and what drives specialization under stable conditions (e.g. resource partitioning promotes species coexistence)

Line 72-74: This seems like an unfinished sentence/typo here. I suggest rewording this to something like “For instance, previous studies have shown that gulls shift to a primarily fish-based [18–20], feeding both in the natural environment and on fisheries discards [21], when natural fish prey abundance increases in their surrounding environment.”

Line 85: capitalize G. There is not much evidence for feeding of black backs at landfills, from what I’ve read in literature and none of these citations seem to suggest that.

Line 122: great black-back gulls eat eggs and chicks opportunistically, but they don’t go on foraging trips to seek out these food items, like they do with fish. I think this would be a good clarification to make here.

Line 128: I don’t think this statement about anesthesia is necessary here

Line 149: I am not following your logic here. You can make this statement about changes in movement reflecting prey availability only if the birds maintain the same breeding stage throughout the low and high capelin periods. It seems that you tracked capelin over a two month period (July-August), in which case birds that were incubating upon initial tagging would have chicks within this July-August window and thus may change their diet based on changes in breeding stage (strong evidence in literature for this).

Line 158: I am not following this. You mention GPS spatial error but then account for it using a time limitation for trips…if you are worried about GPS spatial error you could put a 20 meter buffer around the colony and classify any points within the buffer as “on island” in addition to the 2-hour time foraging trip threshold you defined.

Line 165: perhaps a simple significance test here to justify using means instead of maximum

Line 168- How did you know the birds were foraging? They could have been off the nest loafing for 2 hours near a sandbar?

Line 171: Include the version of R used

Line 173: When you calculate the 50% UD for the group, did you normalize across individuals so that one individual with a longer tag deployment received the same weight as a bird with a shorter tag deployment? Similarly, when you calculate 50% for the individual, you could also normalize trip length to weight all trips equally.

Line 179: be consistent throughout the paper when using “prey period” vs. “capelin availability period”

- Did you control for the family-wise error rate such as using a Bonferroni correction?

- Would be helpful to specify your alpha value somewhere (I am assuming you used alpha = 0.05)

Line 206: It would give you more statistical power to include sex as a random effect, rather than a fixed effect, particularly because this paper is not interested in the effect of sex on foraging.

Line 214: How did you did discriminate between foraging points and transit points on each foraging trip? This seems critical to your quantification of foraging points within the marine environment.

Line 235: Is information on chicks and reproductive success here significant to the study? If it isn’t I would consider removing this information so as to not confuse readers as to the questions of this paper.

Line 281: According to your results table, capelin availability alone was not a significant predictor in the individual model, only the interaction between individual and capelin availability (it also seems like individual is the main driver of the individual:capelin predictor having an effect on response variables). I would be clearer here in the presentation of your results

Line 285: I suggest including a p-value where you mention significance

Table 2: Including sex here adds a level of confusion, since you weren’t interested in sex. I recommend changing sex to a random variable in your models.

Discussion: Why would the same colony have both contextual and non-contextual specialists? I think it would be worth throwing some ideas in here regarding this question (if it is competition driven it suggests something is a limiting resource, but are you suggesting some individuals can outcompete within the same species?)

Reviewer #2: Title

OK.

Abstract

Very well written.

Keywords

OK.

Introduction

Very well written.

Materials and methods

Well written.

The only thing which may be improved (but probably in further protocols) is the number of specimens studied. From my perspective, reaching at least 10-15 specimens in every group should be considered.

Please add information on R version used.

Results

Well written.

Specific comment for the fig. 2: I think the grey scale would be better and ease the interpretation.

Discussion

Well written.

References

Well chosen.

6. PLOS authors have the option to publish the peer review history of their article (what does this mean?). If published, this will include your full peer review and any attached files.

Reviewer #1: No

Reviewer #2: No

---

## [Author Response · Author response to Decision Letter 0]

25 Jul 2021

Respones to reviewers

Reviewer #1: 

This paper presents a study investigating a unique idea in theoretical ecology and is worthy of publication upon addressing suggested edits. The largest overall comment to this paper is there is no clear link made between capelin biomass and great black-backed gull foraging/diet. Are the authors suggesting that black-backs are eating the capelin? Are they suggesting that increase capelin abundance also increases the abundance of larger fish that the black-backs eat? Creating a clearer picture here would be extremely beneficial to the paper.

We added a sentence in the introduction (lines 95-98) and in the discussion (lines 340-341;lines 349-352) to clarify that great black-backed gulls (GBBG) are known to feed on capelin in other coastal regions of Newfoundland. However, since we did not simultaneously perform a diet study, we cannot confirm that the diet was comprised of capelin during high capelin availability period and as such have limited our discussion on the matter to those sentences.

Line by line comments:

Line 66: change “should” to “may”

We have made the suggested change (line 66).

Line 67: it would be beneficial to expand here as to what you mean by “stable” conditions and what drives specialization under stable conditions (e.g. resource partitioning promotes species coexistence)

We added a sentence and defined “stable” conditions. The sentence now states (lines 66-70):

“Specialization, however, can be contextual, whereby an individual may be a specialist under stable conditions (e.g., consistent prey availability), allowing resource partitioning and promoting species coexistence, but shifts to a generalist strategy when conditions change (e.g., released from competition, increased prey availability).”

Line 72-74: This seems like an unfinished sentence/typo here. I suggest rewording this to something like “For instance, previous studies have shown that gulls shift to a primarily fish-based [18–20], feeding both in the natural environment and on fisheries discards [21], when natural fish prey abundance increases in their surrounding environment.”

We adjusted the sentence according to recommendations (lines 73-76).

Line 85: capitalize G. There is not much evidence for feeding of black backs at landfills, from what I’ve read in literature and none of these citations seem to suggest that.

We have updated the previous Good reference from the 1990s to the current Birds of the World (2020) which mentions the use of landfills.

17. Good, TP. Great black-backed gull (Larus marinus), version 1.0. In: SM Billerman, editors. Birds of the World. Cornell Lab of Ornithology, Ithica, New York; 2020. doi:10.2173/bow.gbbgul.01.

“Obtains human refuse by following garbage scows, by roosting at landfills, or by waiting downstream of sewage outfalls (Bent 1921, TPG). Observed at garbage dumps more in winter (Wells 1994) than during breeding season (Verbeek 1979a). There is confusion over the role of “refuse”-fishery-generated waste and garbage from dumps-in diet. Fisheries waste (bycatch and fish offal) is higher in food quality than human food scraps from garbage dumps (Pierotti and Annett 1987). Landfills appear less important to this species than to Herring Gulls (but see Burger 1988a), particularly during breeding (Mudge and Ferns 1982). Thought to be less affected by closing of landfills than is Herring Gull (Buckley and Buckley 1984b, Cavanagh 1992). In Newfoundland, feeds no garbage to chicks (Pierotti 1979); on Appledore I., ME, pairs that fed chicks garbage had reduced breeding success (TPG).”

Line 122: great black-back gulls eat eggs and chicks opportunistically, but they don’t go on foraging trips to seek out these food items, like they do with fish. I think this would be a good clarification to make here.

We argue that GBBG may leave their colony to forage at another seabird breeding island (i.e. foraging trip), depending on the abundances and types of seabirds resident on their breeding island. For instance, we collected observation data on GBBG predation on the South Cabot Island Murre colony (an island nearby our GBBG study colony island), and some individuals appeared to consistently use murres at this colony as a food source. Because they leave the island on which they breed, these trips away from the island would qualify as foraging trips. 

However, we understand the comment and we clarified the sentence by removing ‘localized areas of high density prey’ not to imply that the colony is the primary food source for all GBBGs but a potential food source (lines 122-126). We also included two sentences (lines 126-129) on absence of seabirds other than herring gulls breeding in high number on North Cabot Island.

Line 128: I don’t think this statement about anesthesia is necessary here

This was added according to PlosOne Submission Guidelines. This sentence has been removed and integrated with the ethical statement at the end of the methods (lines 226-233).

Line 149: I am not following your logic here. You can make this statement about changes in movement reflecting prey availability only if the birds maintain the same breeding stage throughout the low and high capelin periods. It seems that you tracked capelin over a two month period (July-August), in which case birds that were incubating upon initial tagging would have chicks within this July-August window and thus may change their diet based on changes in breeding stage (strong evidence in literature for this).

We thank Reviewer 1 for this careful comment. We monitored breeding status throughout the tracking period to account for variation in foraging behaviour related to breeding stage/status. We added a sentence clarifying this at the end of the sentence on line 148-150. You can also see our effort in monitoring breeding stage at lines 131-134 and our sensitivity analysis at lines 244-248. Since most of the individuals failed to breed within the low prey availability period (i.e. prior to capelin spawning), however, we did not include breeding status/stage in the model.

Line 158: I am not following this. You mention GPS spatial error but then account for it using a time limitation for trips…if you are worried about GPS spatial error you could put a 20 meter buffer around the colony and classify any points within the buffer as “on island” in addition to the 2-hour time foraging trip threshold you defined.

We agree with the reviewer and revisited our methods and the reason for the temporal threshold. We clarified our reasons (lines 156-158) for the temporal threshold was not to account for the spatial error of the GPS devices but rather to distinguish between trips away from the island that were likely foraging trips (> 2hr) relative to those were not likely foraging trips (< 2 hr). This temporal threshold has been used in other studies on gulls such as :

34. Maynard LD, Davoren GK. Sea ice influence habitat type use by great black-backed gulls (Larus marinus) in coastal Newfoundland, Canada. Waterbirds. 2018;41: 449–456.

48. Shamoun-Baranes J, Bouten W, Camphuysen CJ, Baaij E. Riding the tide: Intriguing observations of gulls resting at sea during breeding. Ibis. 2011;153: 411–415. doi:10.1111/j.1474-919

62. Camphuysen KCJ, Shamoun-Baranes J, van Loon EE, Bouten W. Sexually distinct foraging strategies in an omnivorous seabird. Mar Biol. 2015;162: 1417–28.

Isaksson N, Evans TJ, Shamoun-Baranes J, Åkesson S. Land or sea? Foraging area choice during breeding by an omnivorous gull. Mov Eco. 2016. 4:11. doi:10.1186/s40462-016-0078-5

Line 165: perhaps a simple significance test here to justify using means instead of maximum

We provided more information on how we chose the model with mean distances parameters (i.e. by comparing the AIC of the mean and maximum models; lines 164-167).

Line 168- How did you know the birds were foraging? They could have been off the nest loafing for 2 hours near a sandbar?

We added a sentence (see lines 167-168) to outline that we cannot distinguish between foraging and roosting at the interval of GPS recording (15 minutes). We have also modified another sentence (see lines 165-167) to clarify that we are measuring habitat use and not foraging per se.

Line 171: Include the version of R used

We added the information required (line 171).

Line 173: When you calculate the 50% UD for the group, did you normalize across individuals so that one individual with a longer tag deployment received the same weight as a bird with a shorter tag deployment? Similarly, when you calculate 50% for the individual, you could also normalize trip length to weight all trips equally.

For tag-deployment, we did a sensitivity analysis to evaluate the impact of this variation in tracking duration among individuals on our results. We calculated the UDs for the first ten days (when all individuals were tracked simultaneously) and for days 10-20 and compared with the UDs across the entire summer. The core foraging areas were at the same locations in all cases, but were smaller when fewer days were used (i.e. first 10 days, days 10-20). We believe our results are therefore robust to tag deployment duration and we did not need to normalize across individuals for the group-level model. We added a sentence at line 247 to clarify our methods.

For foraging trip duration, we reanalyzed the data to normalize by capelin availability period instead of by foraging trip. To normalize UDs at the individual scale, we randomly sub-sampled GPS locations outside the colony so that each individual would have the same number of locations in each capelin availability period. For example, LMG01 had 800 locations during high capelin availability but 1200 during low capelin availability, so we sub-sampled locations during low capelin availability to match 800 locations during high capelin availability. We did this for each individual separately, since our focus was on within-individual variation. This change did not significantly affect the results. We added a sentence at lines 174-177 and adjusted Table 1 and Figure 2 accordingly.

Line 179: be consistent throughout the paper when using “prey period” vs. “capelin availability period”

- Did you control for the family-wise error rate such as using a Bonferroni correction?

- Would be helpful to specify your alpha value somewhere (I am assuming you used alpha = 0.05)

We have gone through the manuscript carefully and changed ‘prey period’ to ‘capelin availability period’. 

We did not conduct pairwise statistical comparisons of BA, but rather calculated BA as the spatial overlap of all possible foraging trip pairs. The resulting BA values then became the response variables that were compared in two statistical tests. Therefore, we did not conduct a Bonferonni correction (or family-wise error rate), as we did not consider it necessary considering the low number of models and the a priori contrasts. We have modified lines 181-183 to clarify our analysis.

We added the alpha value at line 169.

Line 206: It would give you more statistical power to include sex as a random effect, rather than a fixed effect, particularly because this paper is not interested in the effect of sex on foraging.

We agree with the reviewer and have reanalyzed the data with sex as a random effect (lines 213). It did not affect the results and sex was removed from Table 2.

Line 214: How did you did discriminate between foraging points and transit points on each foraging trip? This seems critical to your quantification of foraging points within the marine environment.

We clarified here that we are talking about foraging/roosting points since we can’t distinguish the two. Please refer to lines 155-156 where we state “Within each trip, we considered point locations under 4 km/h foraging/roosting locations, following [48] and [34].”

Line 235: Is information on chicks and reproductive success here significant to the study? If it isn’t I would consider removing this information so as to not confuse readers as to the questions of this paper.

Given the high number of failed compared with successful breeders in the study, we felt that this was an important factor to make readers aware of. We recognize that providing this information may lead to potential confusion, but we felt it was important to clarify that the majority of our results are on failed breeders given that the study occurs during the breeding season.

Line 281: According to your results table, capelin availability alone was not a significant predictor in the individual model, only the interaction between individual and capelin availability (it also seems like individual is the main driver of the individual:capelin predictor having an effect on response variables). I would be clearer here in the presentation of your results

We have added a sentence to clarify that the interaction between individual and capelin availability periods was significant along with the meaning of the interaction (lines 292-294).

Line 285: I suggest including a p-value where you mention significance

We added the range of non-significant p-values considering this is a result for many contrasts which have several p-values (lines 299).

Table 2: Including sex here adds a level of confusion, since you weren’t interested in sex. I recommend changing sex to a random variable in your models.

We agree with this recommendation and made the suggested change (line 213 and table 1).

Discussion: Why would the same colony have both contextual and non-contextual specialists? I think it would be worth throwing some ideas in here regarding this question (if it is competition driven it suggests something is a limiting resource, but are you suggesting some individuals can outcompete within the same species?)

We reworked the third paragraph of the Discussion (lines 364-398) to further address this comment. In particular, we interpret the combination of contextual and non-contextual (and generalist) individuals within the same population in the context of competitively dominant and subordinate individuals.

 

Reviewer #2: 

Title

OK.

Thank you.

Abstract

Very well written.

Thank you.

Keywords

OK.

Thank you.

Introduction

Very well written.

Thank you.

Materials and methods

Well written.

The only thing which may be improved (but probably in further protocols) is the number of specimens studied. From my perspective, reaching at least 10-15 specimens in every group should be considered.

Please add information on R version used.

As we were primarily interested in individual-level responses to the seasonal shift in prey availability, we focused on increasing the number of repeated measures per individual, rather than increasing the number of individuals tracked (given limited funding) to maximize our ability to quantify individual-level responses across contexts, following Wolak et al. (2012).

We added the R version at line 171.

Results

Well written.

Specific comment for the fig. 2: I think the grey scale would be better and ease the interpretation.

We thank the reviewer for this comment, however, since the colours on the map follow to the different figures within the manuscript, we decided to keep the colours. However, we improved the figure by making the land a lighter grey, adding contour lines to the utilization distribution and adding a legend within the map to increase contrasts and ease interpretation.

Discussion

Well written.

Thank you.

References

Well chosen.

Thank you.

---

## [Decision Letter · Decision Letter 1]

5 Oct 2021

Different individual-level responses of great black-backed gulls (Larus marinus) to shifting local prey availability

PONE-D-21-15967R1

Dear Dr. Maynard,

We’re pleased to inform you that your manuscript has been judged scientifically suitable for publication and will be formally accepted for publication once it meets all outstanding technical requirements.

Kind regards,

Vitor Hugo Rodrigues Paiva, Ph.D.

Academic Editor

PLOS ONE

Additional Editor Comments (optional):

Reviewers' comments:

Reviewer's Responses to Questions

**Comments to the Author**

1. If the authors have adequately addressed your comments raised in a previous round of review and you feel that this manuscript is now acceptable for publication, you may indicate that here to bypass the “Comments to the Author” section, enter your conflict of interest statement in the “Confidential to Editor” section, and submit your "Accept" recommendation.

Reviewer #1: All comments have been addressed

Reviewer #2: All comments have been addressed

2. Is the manuscript technically sound, and do the data support the conclusions?

Reviewer #1: Yes

Reviewer #2: Yes

3. Has the statistical analysis been performed appropriately and rigorously? 

Reviewer #1: Yes

Reviewer #2: Yes

4. Have the authors made all data underlying the findings in their manuscript fully available?

Reviewer #1: Yes

Reviewer #2: Yes

5. Is the manuscript presented in an intelligible fashion and written in standard English?

Reviewer #1: Yes

Reviewer #2: Yes

6. Review Comments to the Author

Reviewer #1: The author adequately addressed all comments provided to the initial submission, all parts of the manuscript that were initially unclear were clarified and recommendations provided by myself were carefully considered.

Reviewer #2: I think the paper is very interesting and of very good quality. Thank you for implementing/discussing my suggestions.

7. PLOS authors have the option to publish the peer review history of their article (what does this mean?). If published, this will include your full peer review and any attached files.

Reviewer #1: No

Reviewer #2: No

---

## [Editor Report · Acceptance letter]

11 Oct 2021

PONE-D-21-15967R1 

Different individual-level responses of great black-backed gulls (*Larus marinus*) to shifting local prey availability 

Dear Dr. Maynard:

I'm pleased to inform you that your manuscript has been deemed suitable for publication in PLOS ONE. Congratulations! Your manuscript is now with our production department. 

Kind regards, 

on behalf of

Dr. Vitor Hugo Rodrigues Paiva 

Academic Editor

PLOS ONE